# A fast and sensitive room-temperature graphene nanomechanical bolometer

Andrew Blaikie[1,2,3], David Miller [1,2,3] & Benjamín J. Alemán[1,2,3,4]*

Bolometers are a powerful means of detecting light. Emerging applications demand that bolometers work at room temperature, while maintaining high speed and sensitivity, properties which are inherently limited by the heat capacity of the detector. To this end, graphene has generated interest, because it has the lowest mass per unit area of any material, while also possessing extreme thermal stability and an unmatched spectral absorbance. Yet, due to its weakly temperature-dependent electrical resistivity, graphene has failed to challenge the state-of-the-art at room temperature. Here, in a departure from conventional bolometry, we use a graphene nanoelectromechanical system to detect light via resonant sensing. In our approach, absorbed light heats and thermally tensions a suspended graphene resonator, thereby shifting its resonant frequency. Using the resonant frequency as a readout for photodetection, we achieve a room-temperature noise-equivalent power ($2\,\mathrm{pW\,Hz^{-1/2}}$) and bandwidth (from 10 kHz up to 1.3 MHz), challenging the state-of-the-art.

[1] Department of Physics, University of Oregon, Eugene, Oregon 97403, USA. [2] Materials Science Institute, University of Oregon, Eugene, Oregon 97403, USA. [3] Center for Optical, Molecular, and Quantum Science, University of Oregon, Eugene, Oregon 97403, USA. [4] Phil and Penny Knight Campus for Accelerating Scientific Impact, University of Oregon, Eugene, Oregon 97403, USA. *email: baleman@uoregon.edu

The bolometer is an essential tool used to detect massive energetic particles and electromagnetic radiation. A primary benefit of the bolometer is its ability to detect light deep into the infrared[1], an ability that has advanced thermal imaging, night vision, infrared spectroscopy, and observational astronomy[2]. Emerging applications[3] in scientific and thermal imaging, remote environmental monitoring, THz communication, solar probes, and terahertz communication coupled with the need for increased portability demand that future bolometers work at room temperature and push the limits of speed (bandwidth, BW) and sensitivity (i.e., noise-equivalent power), while also maintaining a large spectral BW. A common method to modify the speed and sensitivity is to change the thermal resistance between the bolometer and its environment[1]. However, both the speed and sensitivity are inversely proportional to the thermal resistance, so a sensitive bolometer is often slow.

The speed-sensitivity trade-off can be evaded by decreasing the bolometer heat capacity, as the speed is also inversely proportional to heat capacity. Being just one atom in thickness, graphene offers a tantalizing prospect for ultrasensitive and ultrafast bolometry[4], because it has the lowest possible heat capacity per unit area of any material. Moreover, graphene possesses an ultra-broadband spectral absorbance[5,6] and is thermally stable up to at least 2600 K (ref. [7]), so a graphene bolometer could detect electromagnetic radiation of nearly any wavelength and withstand high operating temperatures. However, graphene has performed poorly in conventional bolometry[8]—where the electrical resistance serves as the readout for absorbed power—because its electrical resistivity is relatively insensitive to temperature[9]. Although graphene has shown promise in hot-electron bolometry[10–14], in which a weak electron–phonon interaction generates a thermally insulated electron gas with a low electronic heat capacity, these implementations require cryogenic temperatures and lack portability.

Here we pursue an alternative to electrical bolometry and develop a graphene nanomechanical bolometer (GNB). In nanomechanical bolometry[15], absorbed power is measured by monitoring changes in a miniaturized mechanical structure, such as the deflection of a microbeam[16]. In our GNB (illustrated in Fig. 1a), we measure the resonance frequency of a mechanical resonator[15] comprising a suspended graphene membrane (Fig. 1b, c) (see Supplementary Fig. 2 for gallery of all GNBs characterized). Upon absorbing light, the membrane's temperature increases and the resulting thermomechanical stress shifts the resonance frequency[17,18] by an amount

$$\Delta f_0 = \frac{\alpha Y f_0}{2\,\sigma_0(1-\nu)}\Delta T \qquad (1)$$

where $\alpha$ is the thermal expansion coefficient, $\nu$ is the Poisson ratio, $\sigma_0$ is the initial in-plane stress, $Y$ is the two-dimensional (2D) elastic modulus, $f_0$ is the initial frequency, and $\Delta T$ is the temperature change (see Supplementary Note 8 for thermal modeling details). For typical graphene nanomechanical resonators, a $\Delta T \sim 100$ mK will shift the frequency by a full linewidth, a sizable amount that is readily measured. For a given absorbed power ($P_{abs}$), the $\Delta T$ is amplified by the thermal resistance ($R_T$), as determined by Fourier's law $\Delta T = P_{abs} R_T$. The $R_T$ of suspended graphene is abnormally large[19], but to enhance $R_T$ further, we patterned the suspended graphene[20] into a trampoline geometry[21,22] with narrow, tapered tethers (see Fig. 1c). This configuration lets us use the low $C$ of graphene and increase $R_T$, while providing an effective and sensitive means to measure the absorbed light.

## Results

### Description of the GNB fabrication and mechanical measurements.
In our GNB, light is detected by tracking changes to the fundamental mode frequency of a graphene nanomechanical resonator (see Supplementary Methods). The graphene structures are made by transferring graphene[23] onto a silicon/silicon oxide support substrate with patterned holes, resulting in circular drumhead resonators (Fig. 1b). Some drumheads are patterned into trampoline geometries using a focused ion beam (FIB) technique[20] (see Methods for details), as shown in Fig. 1c. We drive motion of the graphene resonators[24] by applying an a.c. voltage between the graphene and the backgate (Fig. 1a), and we measure the motion with a scanning laser interferometer[25] operated with a low-power, power-locked laser. By sweeping the a.c. drive frequency, we obtain amplitude and phase spectra, as seen in Fig. 1d for the first fundamental mode of a trampoline. The resonance frequency can be inferred from either the phase or the amplitude spectrum, which from Fig. 1d is ~10.7 MHz. We obtain the resonance gate dependence by applying a d.c. bias to the graphene, while measuring the amplitude spectrum, as shown in Fig. 1e. By using an electromechanical model (see Supplementary Note 7), the gate dependence (Supplementary Fig. 7) reveals the graphene membrane mass density ($\rho$), Young's modulus ($Y$), and initial stress ($\sigma_0$). We track the frequency during light illumination with frequency modulation detection[26], which uses a phase-locked loop (PLL) with the phase locked on resonance. A key advantage of using frequency modulation is that the GNB response BW is not determined by the resonance linewidth, as it is with amplitude modulation detection. The PLL BW allows tracking up to ~50 kHz. For frequency-shift measurements, we maximize the signal-to-noise ratio in several ways. First, we use the scanning interferometer to obtain a 2D spatial map of the vibrational amplitude of the resonator. A map for a trampoline (Fig. 1f) shows 90° rotational symmetry in agreement with the trampoline geometry and goes to zero near the clamping of the tethers, indicating they are the only point of contact to the substrate. Using these maps, we position the interferometer laser to maximize the amplitude signal. Moreover, we adjusted the a.c. voltage level to just below the onset of bistability to maximize the resonator amplitude and to avoid nonlinear effects, such as phase instability, which can disrupt the phase locking.

### Measurement of the noise-equivalent power.
The noise-equivalent power (pW Hz$^{-1/2}$) of the GNB is calculated with the expression $\eta = \sigma_f \sqrt{t}/(f_0\,R_f)$, where $\sigma_f$ is the frequency noise, $t$ is the measurement time, and $R_f$ is the frequency-shift responsivity (i.e., the fractional change in resonance frequency per unit of absorbed power), defined as $R_f \equiv \frac{1}{f_0}\frac{df_0}{dP_{abs}}$. To determine $R_f$, we illuminate the GNB membrane with an amplitude-modulated heating laser (532 nm) and measure $f_0$ with a PLL. A time recording of $f_0$ when the GNB is exposed to sinusoidally modulated light is shown in Fig. 2a, in which $P_{abs} = 4.4$ nW. Here we assume the absorption is 2.3% of the incident power[6,10,12]. The shift $\Delta f_0$ is inferred from a sine fit (Fig. 2a black curve) as the peak-to-peak amplitude. For the data shown in Fig. 2a, $\Delta f_0 = 8.5$ kHz, corresponding to ~72% of the resonator linewidth. The power dependence of $\Delta f_0$ for a trampoline GNB (Fig. 2b) shows that $\Delta f_0$ is linear with $P_{abs}$ (in the range of 1–100 nW) and therefore $\frac{df_0}{dP_{abs}} = \frac{\Delta f_0}{P_{abs}}$ is a constant. The linear power dependence of $f_0$ was observed in all GNB devices. In Fig. 2c, we plot $R_f = \frac{1}{f_0}\frac{\Delta f_0}{P_{abs}}$ vs. tether width ($w$) for nine different trampolines and three different drumheads; for drumheads, $w$ is given by one-fourth the circumference. The trampoline width ($w$) is indicated in Fig. 3b. We tested trampoline GNBs with $w$ ranging from 200 nm to

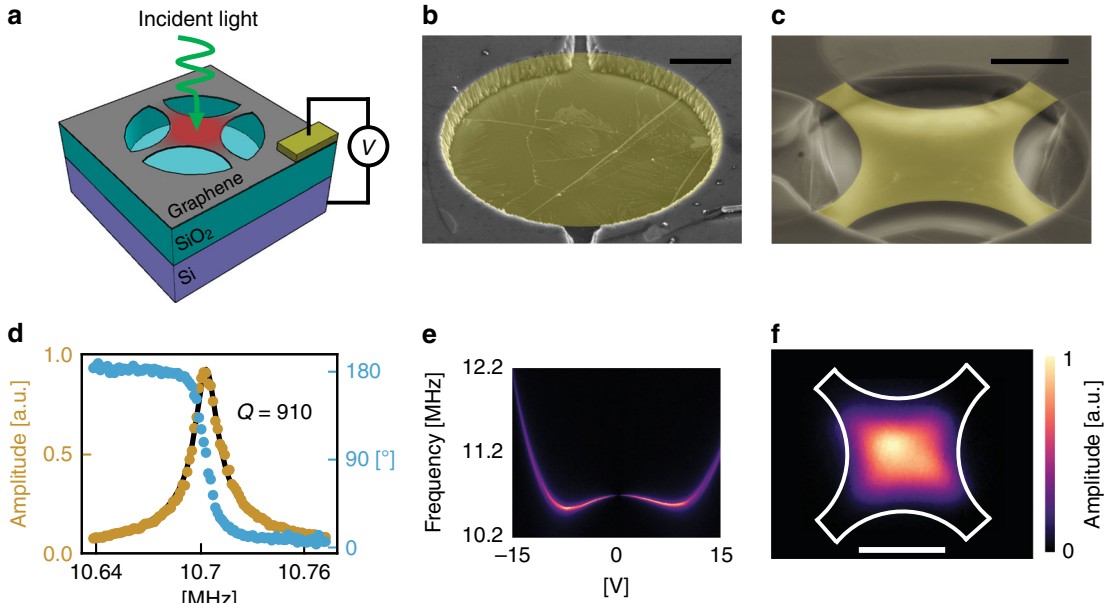

**Fig. 1** Design, images, and mechanical properties of graphene resonators. **a** Illustration of the bolometric detection scheme. A driving voltage, $V_{AC}$, is used to actuate motion and a bias voltage, $V_{DC}$ is used to apply additional tension. The total voltage drop between the suspended graphene and the Si$^{++}$ substrate is $V = V_{DC} + V_{AC}$. Absorbed light tightens the graphene, shifting the mechanical resonance. **b** False-color scanning electron microscope image of a suspended graphene drumhead and **c** trampoline. Regions of collapsed graphene from the focused ion beam cutting process can be seen around the edges of the cavity. Black scale bars are 2 μm. **d** Amplitude-frequency response curve at $V_{DC} = 0.25$ V. The frequency of $V_{AC}$ swept as the mechanical amplitude response is measured. A best fit for a damped driven oscillator is used to calculate the resonance frequency and quality factor. Quality factor is calculated to be $Q = 910$. **e** Amplitude-frequency spectrogram vs. applied d.c. gate bias. **f** Measured mechanical mode shape of a graphene trampoline. Fast steering mirrors were used to scan the probe laser across the device with diffraction limited resolution. The white lines outline the physical device shape as calibrated from a scanning electron microscope image. Scale bar is 3 μm. Color scale is shared with **e** and **f**. Source data are provided as a Source Data file

1.4 μm and GNBs with a $d$ of 6 and 8 μm. In general, the drumheads had $R_f$ values about 1% that of trampolines. Our most sensitive device, a 6 μm diameter trampoline with 200 nm wide tethers, had $R_f \sim 300{,}000$ W$^{-1}$, a factor 100 greater than state-of-the-art nanomechanical bolometers[15]. As seen from Fig. 2c, $R_f$ increases with smaller $w$ for trampolines.

As a measure of the fractional noise, $\sigma_f/f_0$, we used the Allan deviation[27], which we calculate (see Supplementary Methods) from temporal recordings of the frequency while the heating laser is turned off (Fig. 2d). Representative Allan deviation data for varying measurement intervals are presented in Fig. 2e. Across the sampling range and for all devices, the Allan deviation was flat, taking on a value of $\sim 10^{-5}$, indicating that $\sigma_f$ is dominated by flicker noise ($1/f$) and not by thermomechanical noise[28]. In this case, the frequency noise is not generally reduced with a larger quality factor[29,30].

Combining $R_f$ and the Allan deviation (measured at 100 Hz), we calculate the noise-equivalent power $\eta$ for each device and plot $\eta$ vs. $w$, shown in Fig. 2f. This data illustrates that $\eta$ decreases with narrower tether width. A trampoline with a tether width of 200 nm exhibited the best power sensitivity, $\eta = 2$ pW Hz$^{-1/2}$ (at 1 kHz BW), which is also the lowest reported value of noise-equivalent power for a room-temperature graphene bolometer to date[12]. The $\eta$ is much larger for drumheads; the largest value ($\eta \sim 1$ nW Hz$^{-1/2}$) is over 200 times greater than the most sensitive trampoline. From these trends, it is clear that reducing the tether width provides a straightforward means to lower, and thus improve, the GNB's $\eta$.

Our measurement of $\eta$ assumes 2.3% absorption. However, cavity effects and surface contaminants could lead to large deviations from 2.3%. Our cavity modeling (see Supplementary Note 5 and Supplementary Fig. 6) predicts that variations in the

absorption are dominated by interference, which changes the overall intensity at the surface of the graphene membrane. For the device geometry used in this work, the intensity, and thus the effective absorption, is reduced to ~0.6%. Moreover, photothermal back-action cavity effects have a negligible effect on $R_f$ in this configuration (see Supplementary Note 6). For the most sensitive device, cavity effects indicate that the absorbed power could be lower than predicted by the 2.3% absorption estimate and therefore the NEP could be as sensitive as $\eta = 500$ fW Hz$^{-1/2}$. However, by combining the measured frequency shift and resonance frequency gate dependence (Fig. 1e) with predictions from mechanical modeling for $R_f$ (see Supplementary Note 7), we calculate an experimental value for the optical absorption of 2.0%. Surface contaminants on the graphene, which the measured mass density indicates are present, likely increases the total absorption from that predicted from cavity modeling. For the sake of comparison with previous work[10,12], we use the standard absorption estimate[6] of 2.3%.

**Modeling of the frequency responsivity.** The observations of $R_f$ and $\eta$ can be understood through a thermomechanical model that combines a thermal circuit with membrane mechanics. The circuit (shown schematically in Fig. 3a) treats the GNB as a thermal capacitance $C$, given by the membrane heat capacity, in parallel with a thermal resistance $R_T$, governed largely by the tethers (or boundary circumference for drumheads). The absorbed power, $I = P_{abs}$, obeys Fourier's heat law, $\Delta T = P_{abs}R_T$, where $\Delta T$ is the temperature difference between the graphene and the surrounding substrate (assumed to be a room-temperature thermal ground.) By using first-order thermal expansion, we relate $\Delta T$ to the mechanical strain in the GNB membrane to calculate $\Delta f_0$. For

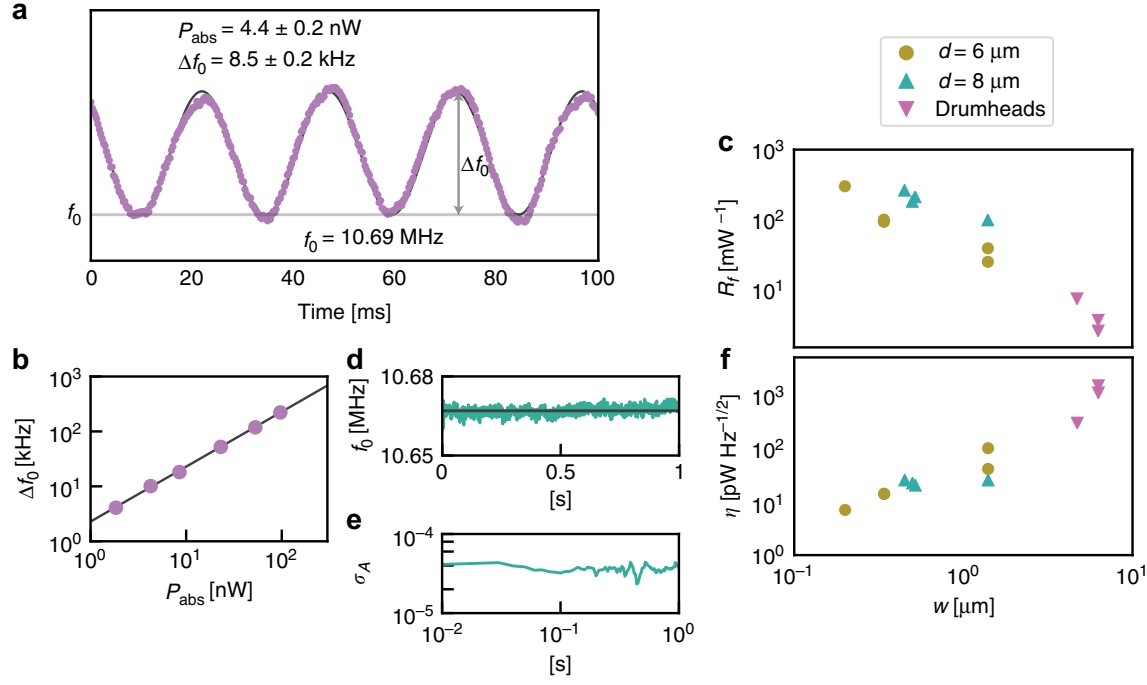

**Fig. 2** Frequency responsivity to absorbed light and frequency noise measurements of graphene resonators. **a** Mechanical resonance frequency vs. time for a 8 μm diameter trampoline with 500 nm wide tethers. The device is subject to 190 nW of incident radiation modulated at 40 Hz. Assuming 2.3% absorption, the absorbed power is $P_{abs} = 4.4$ nW, which causes a frequency shift of $\Delta f_0 = 8.5$ kHz. **b** Measured resonance shift vs. absorbed power. A best-fit line to this data yields a 2.3 kHz nW$^{-1}$ resonance shift per incident power. **c** Frequency responsivity, $R_f$, vs. tether width, $w$, for nine different trampolines and three different drumheads. For the drumheads, the tether width is taken to be 1/4 of the drumhead circumference. **d** Resonance frequency vs. time for a trampoline GNB device. The device is not exposed with heating laser light other than that needed for the measurement. **e** Allan deviation, $\sigma_A$, of the frequency noise vs. measurement time in a log–log plot. The resonance frequency was tracked with the PLL to obtain temporal frequency data. **f** Sensitivity, $\eta$, vs. tether width for nine different trampolines and three different drumheads. Symbol legend is shared between **c** and **f**. Circles indicate a trampoline with a 6 μm diameter, turquoise triangles indicate a trampoline with an 8 μm diameter, and magenta triangles indicate a drumhead resonator of either 6 or 8 μm diameter. Source data are provided as a Source Data file

an absorbed power modulated at angular frequency $\omega$, the model provides an expression for the frequency-shift responsivity

$$R_f(\omega) = -\frac{\alpha Y}{2\sigma_0(1-\nu)}\frac{R_T}{\sqrt{1+\omega^2 R_T^2 C^2}} \qquad (2)$$

again where $\alpha$ is the thermal expansion coefficient, $\nu$ is the Poisson ratio, $\sigma_0$ is the initial in-plane stress, and $Y$ is the 2D elastic modulus. The full details of the thermomechanical model are provided in Supplementary Note 2 and Supplementary Fig. 3. We note Eq. 2 predicts $R_f$ is independent of incident power, in accord with the measurements given in Fig. 2b. In the low-frequency limit (i.e., $\omega \ll \frac{1}{R_T C}$) and with tether resistance $R_T = \frac{\rho_T l}{w}$, where $\rho_T$ is the 2D thermal resistivity of graphene, and $l$ and $w$ are the tether length and width, respectively, Eq. 2 becomes

$$R_f = \frac{\alpha Y \rho_T}{2\,\sigma_0(1-\nu)}\frac{l}{w} \qquad (3)$$

Measurements of $R_f$ vs. $w$ for trampolines given in Fig. 2c agree well with Eq. 3; a fit to $R_f \propto w^{-1}$ for trampolines has a statistical $R$-value of 0.74. Moreover, the model predicts $\eta \propto w$, which is also in agreement with our measurements (Fig. 2f; $R$-value of 0.70). In both cases, the agreement is good, despite some variations in $\sigma_0$ and $l$.

**Measurement of the bandwidth.** Another important metric in a bolometer is the response BW, which determines its ability to detect transient signals and fast variations of the radiation intensity. We characterize the BW in two ways. First, we infer the

BW from the 3 dB roll-off of $R_f(\omega)$, which we get by sweeping the modulation frequency, $\omega$, of the heating laser at fixed power and measuring $\Delta f_0$ with the PLL and a second lock-in (see Supplementary Note 3). We fit the measured $R_f(\omega)$ with Eq. 2 to extract the fit parameter $\tau_T = R_T C$ (i.e., the characteristic time of the circuit), thereby obtaining BW $= \sqrt{3}/(2\pi R_T C)$. An $R_f$ spectrum for a trampoline GNB is illustrated in Fig. 3c, where the black trace is the fit to Eq. 2. This spectrum has a nearly flat response, before falling off at BW ~ 13.8 kHz. As seen from the fit, the measured $R_f(\omega)$ obeys the circuit model very well.

These spectra provide a direct measure of the BW of $R_f$ but are limited by the measurement BW of the PLL. To overcome these speed limitations, our second approach infers the BW from the off-resonant thermomechanical out-of-plane displacement of the graphene membrane, which occurs when thermal stress tightens and locally flattens the membrane[19,31]. In the limit of small displacement and first-order thermal expansion, the mechanical displacement amplitude will be proportional to the change in temperature, $A \propto \Delta T$ (Supplementary Fig. 4). Therefore, the displacement amplitude due to a modulated heating laser of frequency $\omega$ will obey our thermal circuit model and will have the same frequency dependence as $R_f(\omega)$, as given in Eq. 2. For these off-resonant measurements, we sweep the modulation frequency of the heating laser at frequencies well below the mechanical resonance (in the absence of any electrical actuation) and record the amplitude $A(\omega)$ (see Supplementary Note 3). By fitting our measurements of $A(\omega)$ to our model, we extract the thermal response time $\tau_T = R_T C$ and thus BW. Figure 3d illustrates the real and imaginary parts of $A(\omega)$ along with the

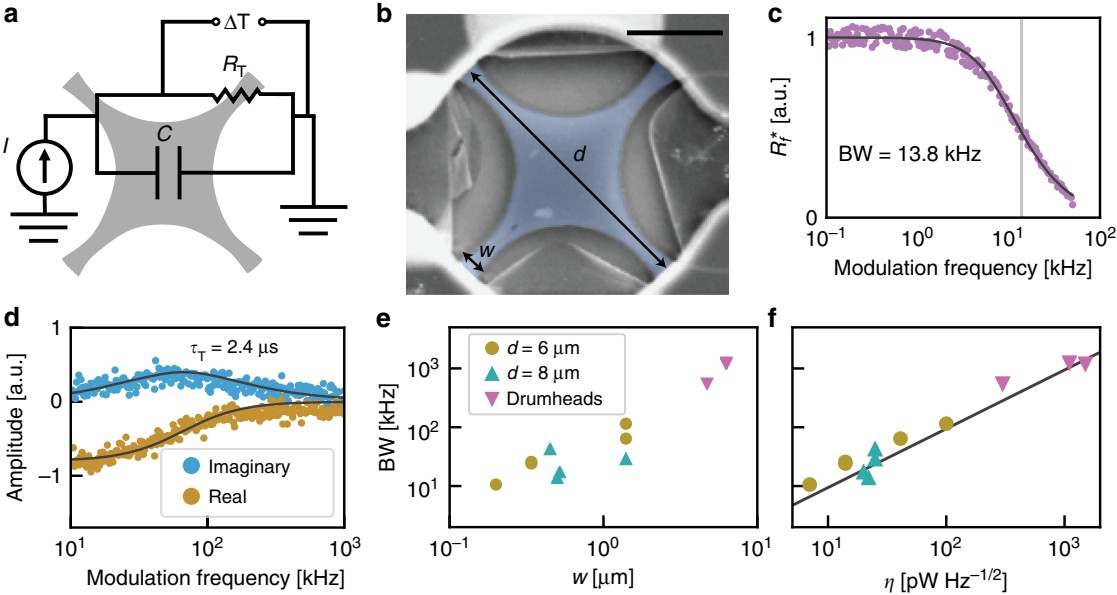

**Fig. 3** Modeling and bandwidth measurements of graphene resonators. **a** Thermal circuit model. **b** False-colored scanning electron microscope image of a trampoline of tether width $w = 200$ nm and diameter $d = 6$ μm. Black scale bar is 2 μm. **c** Normalized frequency-shift responsivity $R_f^*$ as the amplitude of the heating laser is modulated from 100 Hz to 50 kHz. The total resonance shift was found to be constant for low modulation frequency and reached half its maximum value at BW = 13.8 kHz. A thermal circuit model was used to fit the thermal response time of the trampoline; the fitted curve using Eq. 2 is shown in black. **d** Real and imaginary amplitude of thermal expansion induced displacement for a trampoline ($w = 1.2$ μm, $d = 6$ μm). The black curve is a fit to the thermal circuit model. From this fit, we extract the thermal response time, $\tau_T = 2.4$ μs. **e** Bandwidth vs. tether width for nine different trampolines and three different drumheads. For the drumheads, the tether width is taken to be 1/4 of the drumhead circumference. **f** Sensitivity vs. bandwidth for nine different trampolines and three different drumheads. The black line is the linear fit, BW $\propto \eta$ (R-value of 0.97). All bandwidth values in **e** and **f** were inferred from the off-resonant thermomechanical method. Symbol legend and vertical axis is shared between **e** and **f**. Circles indicate a trampoline with a 6 μm diameter, turquoise triangles indicate a trampoline with an 8 μm diameter, and magenta triangles indicate a drumhead resonator of either 6 or 8 μm diameter. Source data are provided as a Source Data file

model fit (black traces) for a trampoline device with $\tau_T = 2.4$ μs or BW = 120 kHz (see Supplementary Fig. 5). Where possible, we compared the BW obtained from $R_f(\omega)$ and $A(\omega)$, finding excellent agreement (see Supplementary Note 3 and Supplementary Table 2). Again, we note that the 3 dB BW is not limited by the mechanical linewidth of the resonator when using frequency modulation[26]. In practice, the BW is limited by either the thermal circuit or PLL BW.

The response BW is strongly correlated with the tether width, where wider tethers produce a faster response. We plot BW vs. $w$ in Fig. 3e. The BW of trampolines ranged between 10 and 100 kHz, while for drumheads the BW was as high as 1.3 MHz. For trampolines, our model predicts

$$\text{BW} = \frac{\sqrt{3}}{2\pi c \rho \rho_T} \frac{w}{l} \frac{1}{A} \qquad (4)$$

where $c$ is the membrane-specific heat, $\rho$ is the membrane mass density, and $A$ is the membrane area. The measured BW data in Fig. 3e agrees well with the model prediction BW∝$w$; for 6 μm diameter trampolines, the linear fit R-value is 0.9. Although our experiments did not broadly sample the device area $A$, our limited data do agree with the prediction BW $\propto A^{-1}$. The BW we measure is likely lower than what we would expect for pristine graphene, as the mass density inferred from the resonance frequency gate dependence (see Supplementary Note 6) is about a factor of ~7.5 greater than pristine graphene.

The BW and the noise-equivalent power are expected to be directly proportional, regardless of the device geometry.

Specifically, our model predicts

$$\text{BW} = \left( \frac{\sqrt{3}}{2\pi} \frac{\alpha_T}{\sigma_A \sqrt{tc\rho}} \frac{1}{A} \right) \cdot \eta \qquad (5)$$

where $\alpha_T = -\frac{\alpha Y}{2 \sigma_0 (1-\nu)}$ is the frequency coefficient of temperature. In Fig. 3f we plot BW vs. $\eta$, as well as a linear fit to the data (black trace), showing an excellent agreement with the prediction BW $\propto \eta$ (R-value of 0.97). In this data, all parameters are constant except for $\sigma_0$ and $A$, but the larger $A$ devices tend to have lower stress $\sigma_0$ so the effects cancel to preserve the linearity. Accordingly, for a given sensitivity, a smaller device area $A$ will boost the speed. Supplementary Note 1 describes the relevant bolometric and mechanical properties of all GNBs characterized with corresponding Supplementary Table 1 and Supplementary Fig. 2.

## Discussion

The response BW and noise-equivalent power demonstrated by our GNB rivals modern high-performance bolometers. Comparing to the sensitivity of previous graphene-based hot-electron bolometers, the GNB is nearly 1000 times more sensitive at room temperature[12]. Assessing the speed and sensitivity together, our lowest $\eta = 2$ pW Hz$^{-1/2}$ compares favorably to the state-of-the-art in room-temperature bolometry, currently set by vanadium oxide and nickel-resistive microbolometers[32–36] ~1–10 pW Hz$^{-1/2}$, while often outperforming the BW of these systems by several orders of magnitude. Moreover, the drumhead GNBs, although not as sensitive, are $10^3$–$10^5$ times faster than modern vanadium oxide bolometers. Using a standard figure-of-merit (FOM)[35]—which assesses the combination of speed and sensitivity

normalized by the device geometry and absorption (see the Supplementary Note 4)—we obtain FOM = $1.18 \times 10^5$ mK ms µm$^2$, where the best reported FOM values for uncooled microbolometers[15,32,33,35] are of the order $10^5$ mK ms µm$^2$. Thus, despite not yet being optimized and a low optical absorption (2.3%), our GNB has already matched these record-low FOM values.

The GNB may enable photodetection applications at very high temperatures because of graphene's extreme thermal stability. Graphene can sustain temperatures up to at least 2600 K[7] and graphene nanomechanical resonators have been shown to operate up to 1200 K[37]. During our experiments, we tested this thermal stability by irradiating the GNB with a laser power of ~400 µW, a level which we estimate would have increased the GNB temperature as $\Delta T = (\Delta f_0/f_0)/\left(R_f/R_T\right)$, (see Supplementary Note 7), and we find $\Delta T = 920$ K or $T \sim 1213$ K, and yet the device remained fully operational and undamaged. Although our experiments used a hybrid electronic/optical scheme for actuation and readout, the GNB can be used in an all-optical platform[20], eliminating the need for on-chip electronics that could degrade at high temperature. Thus, in contrast with most photodetection technologies, our GNB platform is suitable for relevant high-temperature applications, including safety and security applications such as firefighting and industrial process monitoring, and in scientific experiments that take place at high temperature, such as close-proximity solar imaging.

Although we have used the GNB to achieve record bolometric sensitivity at room temperature, it is possible to further improve the noise-equivalent power ($\eta$) of our GNB through practical modifications to material properties and device geometry. Our modeling shows that $\eta \propto \sigma$, where $\sigma$ is stress, so using lower stress graphene[38] would improve $\eta$. Increasing the optical absorption directly improves $\eta$ to incident power. The GNB absorption could be increased to near unity by placing the bolometer in an optimized optical cavity[12,15], at the expense of reduced spectral BW, or by depositing an absorptive material on the GNB, which would also reduce the spectral BW as well as the speed. The simplest way to improve $\eta$ is to reduce the tether width, which can be narrowed down to ~10 nm using FIB (ref. [20]), or to use FIB to create lattice defects in the tethers[39], thereby increasing $R_T$. Taken together, these changes could bring the noise-equivalent power down to the regime of femtowatt sensitivity with 100 Hz response BW.

The fabrication of the GNB used here is scalable and could be used to make dense bolometer arrays. The process used to make GNB devices involves a single-step transfer of chemical vapor deposition graphene on a lithographically defined resonator support frame. Graphene transfer and lithography are both routine processing steps in high-yield, large-scale commercial fabrication. Although FIB is not as scalable as optical lithography, modern FIB, much like e-beam lithography, is used in commercial applications. Therefore, GNB trampolines, which only require a fast, single-pass vector cut, could be made quickly and in large numbers. Drumhead GNBs, while not as sensitive as trampolines, do not require FIB shaping and are routinely fabricated in large arrays[20], and thus could be especially useful for high-speed applications. Although we operated the GNB with a combination electronic actuation and optical readout in this work, our GNB could be fully integrated with on-chip electrical detection and actuation[40,41], allowing it to operate as a stand-alone, packaged technology.

Sensing is an important application of nanoelectromechanical systems (NEMS). In general, the lower the mass of the NEMS device, the more sensitive it will be. By employing low-dimensional materials (e.g., carbon nanotubes and graphene) to operate in the limit of ultralow mass, NEMS sensors have achieved record sensitivity to mass, electrical charge, and force[42–44]. Our GNB uses an ultralow-mass NEMS device to detect light and the GNB's high combination of speed and sensitivity is a direct consequence of its small mass and size. The same frequency-shift sensing mechanism that we use to detect optical power will also inherently respond to mass, charge, and force. Thus, our GNB offers the unique opportunity for multi-mode NEMS sensing, which hybridizes ultrasensitive detection of power with ultrasensitive mass, charge, or force detection. Furthermore, our GNB achieves multi-mode sensing with a single NEMS device and with no further modifications to the device architecture. Using multi-mode sensing, e.g., the GNB could simultaneously detect the mass and energy of an incident particle by detecting the transient frequency shift (from the absorbed kinetic energy) and the steady-state frequency shift (from the added mass). As a consequence of independently measuring the mass and energy of a particle, the GNB would provide a means to measure the momentum of atoms and elementary particles.

In conclusion, we have measured visible light using a graphene nanomechanical resonator by tracking the frequency shifts of the resonator that are induced by light absorption. Using our GNB, we achieve a sensitivity of 2 pW Hz$^{-1/2}$ and a BW up to 1 MHz, thus demonstrating a previously unattainable sensitivity in room-temperature, graphene-based bolometry and greatly outperforming the speed of state-of-the-art room-temperature bolometers. By using graphene, we have demonstrated bolometry in the ultimate lower limit of lattice heat capacity and, consequently, have circumvented the speed-sensitivity trade-off that plagues bolometry. Our GNB fills a vital need in portable medical and thermal imaging, THz spectroscopy, and astronomy for fast, sensitive, and spectrally broadband bolometers and bolometer arrays that operate at and far above room temperature. Furthermore, as the GNB detects power via a nanomechanical mechanism, our work opens the possibility of multi-mode NEMS sensing, which may provide a useful tool in material science, nanoscience, and particle physics to simultaneously measure the energy of a particle along with its electrical charge and mass.

## Methods

**Fabrication of silicon devices**. We fabricated suspended graphene mechanical resonators using standard semiconductor processing techniques. We began by growing 1 µm of wet thermal oxide on Si$^{++}$ wafers at 1100 C. Next, we patterned 6–8 µm diameter holes with AZ1512 photoresist and a direct write laser photolithography system. We etched 600 nm deep into the oxide with a dry inductively coupled plasma etch using a plasma of CHF$_3$ and argon. By leaving some of oxide intact, any collapsed graphene could not cause a short between the suspended graphene and the Si$^{++}$. We then patterned metal electrodes using another AZ1512 direct write photolithography step. Next, we evaporated 5/50 nm Ti/Pt using electron beam evaporation followed acetone liftoff with sonication.

**Semi-dry graphene polymer transfer**. A semi-dry polymer-supported transfer technique was used to place a large sheet of commercial monolayer graphene on Cu foil (Graphenea) over the exposed holes and metal contacts. First, a ~3 µm-thick layer of PMMA A11 was spun onto the Graphene/Cu. The graphene on the backside of the foil was removed with oxygen plasma. Then, a 1 mm-thick piece of polydimethylsiloxane (PDMS) with a ~1 cm diameter hole punched through the middle of it was placed on top of the Graphene/Cu stack. A thin plastic backing was left on the PDMS to increase the rigidity of the film. The Cu foil was etched on a solution of ammonium persulphate (40 mg/ml). The relatively rigid PDMS/PMMA/Gr stack was picked up with tweezers and placed in three sequential water baths before being removed and dried in air. Concurrently, the target substrate with holes was prepared by cleaning it in oxygen plasma before placing it on a hot plate at 155 °C. The now dry PDMS/PMMA/Graphene stack was placed on top of the hot substrate with the through hole covering the entirety of the chip. The substrate was left for ~16 h to improve adhesion between the graphene and the SiO$_2$. The PDMS was then peeled away and the PMMA was removed in flowing Ar and H$_2$ at 350 °C for 3 h. Graphene was scratched off the perimeter of the substrate to prevent shorting to the Si$^{++}$ gate.

**Focused ion beam cutting of trampolines**. We shaped the graphene into trampolines with a focused ion beam. FIB shaping was performed in an FEI Helios 600i

SEM-FIB with a $Ga^+$ source. The ion beam current and voltage were 1.1 pA and 30 kV, respectively. To fabricate a trampoline, four circle outlines were cut into a graphene drumhead using a single beam pass and a dwell time of 1 ms, which was enough to etch a line completely through the suspended graphene sheet. The high tension in the graphene sheet causes the graphene inside the circular cut to pull away from the trampoline resonator and collapse into the cavity. The FIB fabrication technique has a yield of near 100%, with device failures typically due to holes or other defects present in the graphene prior to milling. Although the FIB milling likely induces additional disorder in the graphene sheet, it still maintains its excellent electrical, mechanical, and thermal properties.

**Optical measurements**. Mechanical motion was measured with optical interferometry with a 633 nm HeNe laser. All measurements were performed at room temperature under a vacuum of $P < 10^{-5}$ Torr. Measurement laser powers were kept <1 μW to minimize any heating caused by the measurement. Reflected light was measured with a silicon avalanche photodiode and recorded with a lock-in amplifier. The lock-in amplifier was used to apply an AC voltage to actuate motion in the suspended graphene resonators. A built-in PLL tracked changes to resonant frequencies due to radiation-induced heating or inherent frequency fluctuations. Heating radiation was applied with a 532 nm diode laser modulated with an acousto-optic modulator. The Supplementary Methods provides more detail and Supplementary Fig. 1 gives a schematic of the optical apparatus.

## Data availability

The datasets generated during and/or analyzed during the current study are available from the corresponding author on reasonable request. The source data underlying Figs. 1d–f, 2a–f, 3c–f, and Supplementary Figs. 3b, 5a, b, 7, and Supplementary Table 1 are provided as a Source Data file.

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

## Acknowledgements

We acknowledge the facilities and staff from the Center for Advanced Materials in Oregon (CAMCOR) and the use of the University of Oregon's Rapid Materials Prototyping facility, funded by the Murdock Charitable Trust. We thank Shannon Boettcher, Joshua Ziegler, Rudy Resch, and Kara Zappitelli for scientific discussions and feedback related to this work. This work was supported by the University of Oregon and the National Science Foundation (NSF) under grant number DMR-1532225.

## Author contributions

A.B. and B.J.A. conceived and designed the experiments. D.M. and A.B. fabricated the graphene devices. D.M. and A.B. designed and built optical measurement apparatus. A.B. performed the experiments with assistance from D.M. and A.B. analyzed the data. A.B. and B.J.A. developed the modeling. All authors contributed to writing the manuscript. B.J.A. supervised the work.

## Competing interests

The University of Oregon has filed a U.S. Patent application # 16/297384 which covers the bolometer aspects of the paper. A.B., D.M. and B.J.A are affiliated with this organisation and the patent. The patent application is pending.
