## [Peer Review File · Nature Communications]

Reviewers' comments:

Reviewer #1 (Remarks to the Author):

In this work, Blaikie and coauthors discuss the fabrication and performance of free-standing graphene sheets employed as micro-bolometers. Contrary to typical approach to bolometry, which rely on the measurement of the electrical resistance, they measure the resonance frequencies of suspended membranes. Incident light heats the membrane, this heating results in an increased tension due to the well-know negative thermal expansion coefficient of graphene, producing an increase of the eigenfrequencies. They investigate two geometries, trampoline and drums, which show different thermal resistance to the heat bath (the silicon substrate) allowing them to play with the tradeoff between time response and sensitivity.

As a general comment I like this work. It is a good piece of science, the fabricated geometries are simple but smart, and the authors provided good statistics by fabricating several samples with slightly different geometric parameters. Also, the performance of the sensor are very good. Here the use of graphene really provides an additional value and an improvement with respect to the state-of-the-art, which is something that is not always true.

However, I believe that the performance increase is somehow overestimated if compared to what has been achieved with resistive microbolometer. As an example I found the two works:

(1) 10.1109/IRMMW-THz.2016.7758405

(2) 10.1109/TMTT.2016.2562623

where (1) shows a NEP of 1 pW/sqrt(Hz) and a response time below 1ms, i.e. BW above 1kHz, at 300K, while (2) shows a NEP of 10 pW/sqrt(Hz) and bandwidth above 10kHz. These performacne are somehow in the range of what reported in this work. Contrary to what stated by the authors in the manuscript, it seems to me that resistive bolometers shows performance comparable to the reported devices. The bibliographic reference used to make this comparison are just two and the impression to the reader is that they were just cherry picked to have a more favourable performance gain. Honestly, I believe that the authors just focused on the 'graphene world' when discussing the existing litterature, however, considering its relevance of such comparison, this issue needs to be fixed.

Major comments

* I think that the discussion about the response time is incomplete. The q factor of 900 is not very high on an absolute scale, as an example if compared with microbridge/cantilever resonators. This result in a tradeoff between intensity and bandwidth. As an example for 10nW the expected frequency shift is of about the linewidth (11kHz), which is comparable with the bandwidth of the system (~13kHz). For lower intensity the detection of ΔF requires longer integration time, limiting the time response. The authors should comment about this, if I'm wrong just in the reply.

* It is not clear what the minimum detectable power is. Apparently 1-10mK is the minimal temperature variation, altought this is not very clear in the text. How much power is it? How much is the light absorbtion of the device? What is the absolute sensitivity of the device? If there are technical difficulties to answer such questions, I believe that these issues should at least be discussed.

* It is not clear how they estimated the temperature of the graphene in the high power irradiation measurement. Is light absorbed by the membrane or by the substrate nearby/below?

* The device geometry employed remind me the silicon nitride trampolines used in quantum optomechanics experiments (10.1103/PhysRevLett.116.147202). If the authors got inspiration for the device design form this (or another) work, it would be fair to cite it.

Minor corrections

* The authors says: "The power sensitivity is inversely proportional to the thermal resistance between the bolometer and its environment¹, $1/R_T$, and therefore can be improved by increasing R_T ." But I don't understand this sentence. Or the sensitivity is proportional to R , and thus improves if R increases, or it is inversely proportional to R , and thus improves if R decreases.

* The authors says: "...which has four lobes corresponding to each of the four trampoline tethers...". This sentence is ambiguous, I think the author means that the figure has one lobe and a 90° rotational symmetry, indicating that they are measuring the first mode and also showing the role of the tethers as mechanical boundary conditions.

* R_f is ambiguous (R_t is the Thermal resistance). is it standard notation?

* Figure 1

- Caption (a): the expression "across the device" is ambiguous, it could indicate that the voltage drop is across the membrane.
- (c) the figure quality is quite poor and the x scale is almost useless. I suggest to indicate the ticks each 0.01 MHz.
- The resonance frequency related to Figure 1(c) in the main text is ~ 10.7 , while in the figure is 10.7042. I suggest to be consistent in the precision or just use the most precise value in the text.

* Figure 2

- Set the maximum of y scale of (b) and (c) y to 10^3

* Figure 3

- The authors should use different markers in the plot of the response time to allow the reader to distinguish between the two evaluation procedures they employed.

Reviewer #2 (Remarks to the Author):

The paper by Blaikie et al describes the use of a suspended graphene resonator as a sensitive light detector. The principle is the fact that graphene absorbs light across a wide bandwidth, and when it does, it heats up just a little bit. In response to this, the suspended graphene undergoes tensioning (as graphene has a negative CTE). This results in the change of frequency, which as in all NEMS, is easier to detect using sensitive lock-in techniques.

Overall, the authors present convincing evidence that this is a suitable technique to detect temperature changes. They show that in the best case scenarios, they either get a sensitivity of $2\text{pW}/\text{rt. Hz}$ or have an upper bandwidth of 1.3 MHz (by the way, this needs clarification in the abstract as it is liable to misinterpretation the way it is presented currently - they do not achieve both simultaneously).

The authors correctly point out that for bolometers reported thus far, as the RT goes up, the speed goes down. However in their bolometer I would expect that as Q (quality factor of the resonator) goes up, the speed should also suffer. The entire paper has no treatment of quality factor, a surprising fact considering that almost all the literature in NEMS have focussed on the fact that the relatively high Q factors are a crucial aspect of their astounding sensitivity. Indeed, one would expect that for lower "w" resonators (the trampoline design), the Q would be higher for smaller "w". Have they observed this? Higher Q means that the bandwidth of detection will be reduced in their case (as the Q factor will increase the amount of time between two sensing events), whereas lower Q would mean lower sensitivity, precisely as the authors seem to report.

Indeed the paper has a fairly substantial treatment of the bandwidth limitations in both the main text and the supplementary, but that treatment does not include the Q factor. I urge the authors to look into this a detail and report that in any resubmission. For example, does tensioning affect Q? If so, would that also be part of the sensitivity? More importantly - could such resonators be working in the highly nonlinear regime (graphene resonators are almost exclusively observed with duffing or higher order nonlinearity when driven). How would this affect sensitivity? I appreciate that these measurements are not easy, and teasing out the relative contribution of these effects is not trivial, but a discussion on these limitations of understanding (and if any resolutions can be sought from their data and further analysis) would be helpful.

Also a quick note on notations. They have used "I" for power, which is very confusing, especially as this paper stands alongside NEMS papers where "I" would stand for current. I suggest that the authors use something less easily confused!

I have no idea what the difference between "trampolines" and "drumheads" is. A clear figure and an SEM image might help? Are drumheads without any "tethers"? If so, they are simply fully suspended graphene membranes? In that case their Q should be much lower, I suspect - is this borne out?

Heating is carried out entirely by lasers and the temperature is estimated. Instead, a calibration run with an in-situ hot-plate heater would be very helpful and improve the accuracy of their measurement.

Also - how do we know that the absorption is not occurring in an optical cavity between the graphene and the underlying substrate? After all, some of the light will pass through graphene (I assume monolayer, in which case, depending on the wavelength, almost 90% will pass through at first pass!)

Overall, in principle I support the publication of this work. However, it needs further reasoning and bearing out of facts as above (and perhaps even additional experiments) to ensure that the authors themselves are convinced about the issues raised here - which I suspect might potentially serve to improve their manuscript.

Reviewer #3 (Remarks to the Author):

The authors of the manuscript present an implementation of a graphene nanomechanical resonant bolometer. Whereby, a nanomechanical resonator is used to detect radiation by measuring frequency shifts that result from opto-thermal heating of the graphene structure. The paper demonstrates high sensitivity and/or high bandwidth for the nanomechanical resonators that rivals that of state of the art devices, the paper further gives device fabrication details, some device statistics, and a simplified model to explain the device behavior.

Although the paper is interesting, it simply has not sufficient novelty in terms of science or technology to be accepted for publications in nature communications. If the authors would like to submit again to the same journal, I strongly urge them to expand their study of both the device and the physics behind it before resubmitting.

Knowing that I believe that no cosmetic changes are going to be enough, and a re-submission should only be done after more theoretical and experimental work is performed, below are some points that could increase the depth and breadth of this work:

- Study of the spectral response of the bolometers: there exist an expansive literature on interference effects to which suspended graphene NEMS are subjected. This will affect the response of the device which will certainly not be flat band. A spectral study of the device is therefore necessary.
- Effect of non-coherent wide-spectrum radiation on the response of the device: subjecting the

device to a non-laser light illumination, and studying the effect of multi-spectral illumination on the device is also necessary to claim practical significance.

- The DC voltage can be used to preferentially enhance the bolometer's response to a certain wavelength once the spectral response of the Si/SiOx/vacuum/graphene is taken into account. A detailed experimental study of the effects of DC voltage would therefore be of interest.
- The authors do not attempt to fit the data (plots 2.c, 3.b, and 3.d) to their model, and extract useful parameters. Instead they rely on textbook values in SI.
- The photothermal back-action is well known (Barton et al., Metzger et al.), and should be quantified and modeled by looking at changes in linewidth.
- Most importantly, either the authors do not explain this well enough in the text or they have committed a category mistake: In the model they seem to assume that the thermal time constant is the same as the device response time, i.e. BW. That is incorrect. The thermal time constant is set by the thermal conductivity, capacity, and interface resistance (Dolleman et al.) whereas the response of a resonant bolometer, i.e. its BW, is that of the mechanical linewidth (to a first order approximation), it is not surprising then that the BW measured using the PLL seems to be equal to the mechanical linewidth of the resonator. Thus equating the mechanical and thermal time constant is a fundamental error.
- The authors have completely ignored the mechanics (dynamics) of their resonator in their model, a complete model should account for such effects.

Response to Referees

Reviewer #1:

Major comments

- “As a general comment I like this work. It is a good piece of science, the fabricated geometries are simple but smart, and the authors provided good statistics by fabricating several samples with slightly different geometric parameters. Also, the performance of the sensor are very good. Here the use of graphene really provides an additional value and an improvement with respect to the state-of-the-art, which is something that is not always true.”

We appreciate the positive comments and agree with the reviewer.

- “However, I believe that the performance increase is somehow overestimated if compared to what has been achieved with resistive microbolometer. As an example I found the two works:
(1) 10.1109/IRMMW-THz.2016.7758405
(2) 10.1109/TMTT.2016.2562623
where (1) shows a NEP of 1 pW/sqrt(Hz) and a response time below 1ms, i.e. BW above 1kHz, at 300K, while (2) shows a NEP of 10 pW/sqrt(Hz) and bandwidth above 10kHz. These performance are somehow in the range of what reported in this work. Contrary to what stated by the authors in the manuscript, it seems to me that resistive bolometers shows performance comparable to the reported devices. The bibliographic reference used to make this comparison are just two and the impression to the reader is that they were just cherry picked to have a more favourable performance gain. Honestly, I believe that the authors just focused on the 'graphene world' when discussing the existing litterature, however, considering its relevance of such comparison, this issue needs to be fixed.”

We thank the Reviewer #1 for providing additional references to compare our device to conventional resistive bolometers. We have added more citations and changed our wording to provide a more comprehensive comparison based on the reviewer's feedback. We have also cited the second work referenced by the reviewer (10.1109/TMTT.2016.2562623)⁶, as it is representative of the state-of-the-art in resistive bolometry. The first work referenced by the reviewer (10.1109/IRMMW-THz.2016.7758405) would be representative of the state-of-the-art in resistive bolometry, showing an NEP of 1 pW/Hz^{1/2} and a bandwidth of ~300 Hz. However, we do not reference this work because it is a very brief conference paper with few details of the experiment or analysis, and we could not find the results in any peer-reviewed paper by the same authors. To further improve the comparison to other high-performance bolometers, we now also refer to the figure-of-merit⁷, which provides of a comparison for a very large number of devices, materials, configurations and bolometer types (mechanical, electronic, etc.).

The reviewer's assessment that we were focused on the “graphene world” is correct. Our intention was to stress the improvements shown with our graphene bolometer design over current attempts to take advantage of the near-ideal bolometric properties of graphene.

- “I think that the discussion about the response time is incomplete. The q factor of 900 is not very high on an absolute scale, as an example if compared with microbridge/cantilever resonators. This result in a tradeoff between intensity and bandwidth. As an example for 10nW the expected frequency shift is of about the linewidth (11kHz), which is comparable with the bandwidth of the system (~13kHz). For lower intensity the detection of delta F requires longer integration time, limiting the time response. The authors should comment about this, if I'm wrong just in the reply.”

Reviewer #1's comment on the response time (bandwidth) would be correct if we had employed amplitude modulation (i.e. slope detection). However, we did not use amplitude modulation in our measurements; we used frequency modulation⁸. When considering the tradeoffs between sensitivity and bandwidth, the quality factor does not limit the bandwidth when using frequency modulation. This is because frequency modulation techniques track the resonance frequency of the resonator directly by actively feedbacking back on deviations in phase, unlike the amplitude the resonance frequency responds simultaneously with any external force. In practice, however, the measured bandwidth will be limited by either the thermal response time or the PLL feedback.

We have added text and references to clarify that FM modulation is not limited by the Q-factor.

- “It is not clear what the minimum detectable power is. Apparently 1-10mK is the minimal temperature variation, although this is not very clear in the text. How much power is it? How much is the light absorption of the device? What is the absolute sensitivity of the device? If there are technical difficulties to answer such questions, I believe that these issues should at least be discussed.”

The minimum detectable power is the amount of absorbed power that could be detected above the noise. In the original manuscript, we reported the noise equivalent power (NEP or η), as it is a standard metric. The minimum detectable power is just NEP/\sqrt{t} where t is the measurement time. For our most sensitive device, the minimum detectable power is $NEP/\sqrt{t} \sim 60$ pW and the noise equivalent power is $NEP \sim 2$ pW/Hz^{1/2}.

In the original manuscript, we assumed a device absorption of 2.3%. In the revised manuscript, we model changes to the absorption caused by cavity effects, and also develop a thermomechanical model to independently estimate the absorption. From this modeling, we conclude that 2.3% is an accurate and reasonable estimate. This modeling, which is significantly expanded, is now found mostly in the supplementary information, and key results are presented in the main manuscript.

The minimum temperature variations is only an estimate based on mechanical modeling to motivate the use of resonance-based detection for room temperature graphene bolometry. It estimates the temperature difference between the substrate and the graphene that could be detected using resonant sensing. We have expanded the discussion of minimum detectable temperature to make this clear, and we provide a more detailed modelling of the minimal detectable temperature in the supplementary information.

- “It is not clear how they estimated the temperature of the graphene in the high power irradiation measurement. Is light absorbed by the membrane or by the substrate nearby/below?”

We agree that our discussion of the high-power irradiation in the manuscript was unclear. The details were provided in the original SI, but the manuscript did not clearly point the reader to these details. We have added text to the manuscript that briefly discusses the temperature estimate and point the reader to the SI for a revised and complete description of the thermal circuit and mechanical modelling.

The observed bolometric photodetection is due to the membrane absorbing incident light. Although some light will be absorbed by the substrate, the substrate has a MUCH larger thermal mass compared to the graphene, so the substrate temperature will change only slightly and the heat will be dissipated in the substrate. Thus, the light absorbed by the substrate will have a negligible effect on the graphene membrane temperature and response.

- “The device geometry employed remind me the silicon nitride trampolines used in quantum optomechanics experiments (10.1103/PhysRevLett.116.147202). If the authors got inspiration for the device design form this (or another) work, it would be fair to cite it.”

We employed the trampoline geometry to increase the thermal resistance between the suspended graphene and substrate. Silicon nitride trampolines are fabricated in optomechanics in order to decrease the mechanical damping. So, the reasoning behind the two uses is unrelated. However, due to the similarity between the geometries we added these citations in the manuscript.

Minor corrections

- “The authors says: "The power sensitivity is inversely proportional to the thermal resistance between the bolometer and its environment $1, 1/R_T$, and therefore can be improved by increasing R_T ." But I don't understand this sentence. Or the sensitivity is proportional to R , and thus improves if R increases, or it is inversely proportional to R , and thus improves if R decreases.”

We thank Reviewer #1 for the comment. In sensitivity (i.e. noise-equivalent power) measurements, a lower value of sensitivity is better than a higher value ($1 \text{ pW/Hz}^{1/2}$ is more sensitive than $10 \text{ pW/Hz}^{1/2}$); that is, a device with a lower numeric value of sensitivity is “more sensitive” to whatever it is sensing. In our case, a lower power sensitivity means we can measure smaller changes in power. We believe this subtlety was the source of the reviewer’s confusion. In the manuscript and the supplementary information, we use the words “sensitivity” and “noise equivalent power” interchangeably.

- “The authors says: "...which has four lobes corresponding to each of the four trampoline tethers...". This sentence is ambiguous, I think the author means that the figure has one lobe and a 90° rotational symmetry, indicating that they are measuring the first mode and also showing the role of the tethers as mechanical boundary conditions.”

We thank the reviewer for pointing out the ambiguity in our wording. We have used the reviewers' suggestion to improve the clarity of the wording in the manuscript.

- “R_f is ambiguous (R_t is the Thermal resistance). is it standard notation?”

Yes, R_f is standard notation for frequency responsivity in mechanical bolometry^{9,10}.

- “Figure 1
 - Caption (a): the expression "across the device" is ambiguous, it could indicate that the voltage drop is across the membrane.
 - (c) the figure quality is quite poor and the x scale is almost useless. I suggest to indicate the ticks each 0.01 MHz.
 - The resonance frequency related to Figure 1(c) in the main text is ~10.7, while in the figure is 10.7042. I suggest to be consistent in the precision or just use the most precise value in the text.”

We thank the reviewer for this observation. We have updated Figure 1 accordingly.

- “Figure 2
 - Set the maximum of y scale of (b) and (c) y to 10³”

We thank the reviewer. We have changed the scale of Figure 2. (b) and (c) accordingly.

- “Figure 3
 - The authors should use different markers in the plot of the response time to allow the reader to distinguish between the two evaluation procedures they employed.”

We thank the reviewer for this comment. We have made this clear in the text and standardized the plot to include only plot the bandwidth as estimated from the thermal response time.

Reviewer 2:

- “Overall, the authors present convincing evidence that this is a suitable technique to detect temperature changes. They show that in the best case scenarios, they either get a sensitivity of 2pW/rt. Hz or have an upper bandwidth of 1.3 MHz (by the way, this needs clarification in the abstract as it is liable to misinterpretation the way it is presented currently - they do not achieve both simultaneously).”

We agree with Reviewer #2 that we have provided convincing evidence that our system can detect temperature, although we note the reported application is the detection of absorbed optical power. We thank the reviewer for the comment about the abstract. We agree. We have changed the wording in the abstract to address this potential misinterpretation.

- “The authors correctly point out that for bolometers reported thus far, as the RT goes up, the speed goes down. However in their bolometer I would expect that as Q (quality factor

of the resonator) goes up, the speed should also suffer. The entire paper has no treatment of quality factor, a surprising fact considering that almost all the literature in NEMS have focussed on the fact that the relatively high Q factors are a crucial aspect of their astounding sensitivity. Indeed, one would expect that for lower “w” resonators (the trampoline design), the Q would be higher for smaller “w”. Have they observed this? Higher Q means that the bandwidth of detection will be reduced in their case (as the Q factor will increase the amount of time between two sensing events), whereas lower Q would mean lower sensitivity, precisely as the authors seem to report.”

We thank the reviewer for their comments. We originally limited our discussion of the Q-factor for several reasons:

- 1. Reviewer #2’s comment that the speed would go down as the Q-factor goes up would be correct if we had been employing amplitude modulation (i.e. slope detection), as is sometimes done in NEMS sensing. However, we did not use amplitude modulation in our measurements. We used frequency modulation. One key advantage of frequency modulation is that the bandwidth is not limited by the mechanical linewidth, as discussed by Albrecht et al.⁸ Instead, the upper limit of the bandwidth is determined by the mechanical resonance frequency, although in practice the bandwidth is limited by the PLL feedback or the thermal response time.*
- 2. The NEP (sensitivity) would improve with higher Q (as Reviewer #2 points out), if the NEP was limited by thermomechanical noise since a higher Q lowers the thermomechanical noise as $\eta_{TM} \propto 1/\sqrt{Q}$. However, our NEP is not limited by thermomechanical noise. Instead, our NEP is limited by characteristic fluctuations in resonance frequency, which we have not observed to correlate heavily with quality factor.*

Therefore, in our measurements both the bandwidth and the sensitivity are not directly affected by quality factor. We have added text to the manuscript to clarify that the bandwidth and the NEP are independent of the Q. We have also added the Q factor as a column to Table 1 in the SI, so that it can be compared with other parameters, including the frequency noise. Table 1 in the SI also has the mechanical linewidth (derived from the Q) to compare directly to our bolometer response bandwidth. As can be seen from the quality factor data presented in the SI, the variation in quality factor cannot account for the variation in bandwidth and sensitivity in the bolometers characterized in this work. There are also several instances of the linewidth being substantially smaller than the measured bandwidth.

- “Indeed the paper has a fairly substantial treatment of the bandwidth limitations in both the main text and the supplementary, but that treatment does not include the Q factor. I urge the authors to look into this a detail and report that in any resubmission. For example, does tensioning affect Q? If so, would that also be part of the sensitivity?”

We thank the review for their comments on tensioning. As stated above, the quality factor does not determine the bandwidth in our measurements, so any change to Q due to tension would not

play a role in changing the bandwidth. However, our modeling predicts that increased tension will decrease the bolometer NEP as $NEP \propto \sigma_0$, where σ_0 is the initial in-plane stress (tension). Again, this decrease in sensitivity is due directly to the increased tension and not from a change in quality factor. We measure the stress dependence, and find that NEP does in fact increase with tension; the results are given in the SI. We have added this modeling and a discussion of the Q factor and the stress measurements to the main manuscript.

- “More importantly - could such resonators be working in the highly nonlinear regime (graphene resonators are almost exclusively observed with duffing or higher order nonlinearity when driven). How would this affect sensitivity? I appreciate that these measurements are not easy, and teasing out the relative contribution of these effects is not trivial, but a discussion on these limitations of understanding (and if any resolutions can be sought from their data and further analysis) would be helpful.”

We thank the reviewer noticing this point as other readers might also have this question. Our measurements depend on working in the linear regime, as PLL techniques are complicated with the bi-stability associated with non-linear resonance. We expect that if we were working in the nonlinear regime the frequency fluctuations would be larger due to the bi-stability of the resonance amplitude. Before each measurement we verified that we were not working in the nonlinear regime by taking an amplitude frequency response curve and then sufficiently reduced the driving voltage to ensure we were operating in the linear regime. We have added text to the manuscript to make it clear that we are in the linear regime.

- “Also a quick note on notations. They have used "I" for power, which is very confusing, especially as this paper stands alongside NEMS papers where "I" would stand for current. I suggest that the authors use something less easily confused!”

We thank the reviewer for this comment. We had originally chosen to use “I” because it is analogous to heat current in the thermal “Ohm’s” law, which we employ in our modeling. However, we agree with the reviewer, and we have changed “I” to “ P_{abs} ” for power absorbed in the text to make this variable less confusing. However, we keep the “I” in the figure illustrating the thermal circuit model as P_{abs} is an analog to a current source.

- “I have no idea what the difference between “trampolines” and “drumheads” is. A clear figure and an SEM image might help? Are drumheads without any “tethers”? If so, they are simply fully suspended graphene membranes? In that case their Q should be much lower, I suspect - is this borne out?”

We thank the reviewer for this question. We have added text and an SEM image to the manuscript to clarify the difference between drumheads and trampolines. Also, SEM images of all devices characterized are presented in the SI, and we point out these images in the main. As a general trend yes, the quality factor of drumheads is less than that of trampolines. We have added a reporting of all quality factors in Table 1 in the SI.

- “Heating is carried out entirely by lasers and the temperature is estimated. Instead, a calibration run with an in-situ hot-plate heater would be very helpful and improve the accuracy of their measurement.”

We thank the reviewer for the comment. In this work, the main finding is the detection of light using a nanomechanical bolometer. While we estimate the minimal detectable temperature and the temperature of the device when operated at high-power, our measurements of the NEP do not require a calibration of temperature, but only require a calibration of the incident optical power, which we perform and report in the methods. The temperature sensitivity of the mechanical resonance is estimated using mechanical modeling and is only calculated in order to motivate mechanical bolometry. Therefore, a temperature calibration would not produce a result immediately useful for this manuscript.

In-situ heating of the sample would increase the temperature of both the substrate (thermal reservoir) and the suspended graphene, convoluting the thermal expansion of the substrate and graphene. In bolometry, only the suspended graphene is substantially heated, while the substrate remains at a fixed temperature. Therefore, this measurement would not provide an accurate calibration.

- Also - how do we know that the absorption is not occurring in an optical cavity between the graphene and the underlying substrate? After all, some of the light will pass through graphene (I assume monolayer, in which case, depending on the wavelength, almost 90% will pass through at first pass!)”

We agree with the reviewer, and we have added a careful analysis of cavity effects, including an experimental measurement of the cavity depth and the oxide thickness, to the SI. We find cavity effects are minimal because of graphene’s very low reflectivity ($< 0.1\%$). Nevertheless, our model shows that the cavity is most likely reducing the absorption, instead of increasing absorption. Therefore, our NEP (sensitivity) measurements are slightly conservative.

- “Overall, in principle I support the publication of this work. However, it needs further reasoning and bearing out of facts as above (and perhaps even additional experiments) to ensure that the authors themselves are convinced about the issues raised here - which I suspect might potentially serve to improve their manuscript.”

We thank the reviewer for their support in publishing our work. We believe that we have addressed all issues raised by the reviewer, and we look forward to a positive response.

- **Reviewer 3:**

“Although the paper is interesting, it simply has not sufficient novelty in terms of science or technology to be accepted for publications in nature communications. If the authors would like to submit again to the same journal, I strongly urge them to expand their study of both the device and the physics behind it before resubmitting. Knowing that I believe that no cosmetic changes are going to be enough, and a re-submission should only be

done after more theoretical and experimental work is performed, below are some points that could increase the depth and breadth of this work:”

We respectfully disagree with the reviewer regarding the novelty of our work. We encourage the reviewer to read our summary of the significance of our work provided above to the editors. However, we thank reviewer #3 for their comment and we have made changes to the manuscript to highlight the many novel aspects of the work. Moreover, we do agree that a more complete discussion of the physics of our approach should be in the main manuscript and we should include more measured values. We have expanded our study in accordance with this feedback in the following ways:

- 1. First, we add a resonance frequency vs. DC gate voltage measurement and use this data to perform a three-parameter fit to extract the initial stress, mass density, and elastic modulus of the suspended graphene. These fitted mechanical values are used in a thermomechanical model to independently predict and calculate the graphene heat capacity, thermal resistance, graphene optical absorption, and the frequency responsivity of bolometer. All these independent measurements agree with the literature. The frequency responsivity agrees (within 10%) with the independently measured value. This mechanical modeling adds significant theoretical backing to this work.*
 - 2. Second, we measured the cavity parameters directly with atomic force microscopy and used these measurements to perform a theoretical analysis of cavity effects on the absorption of the graphene membrane. We discuss how the cavity parameters can be used to enhance the absorption and how the cavity affects the spectral bandwidth.*
 - 3. Third, we have estimated the strength of photothermal back-action as it relates to nanomechanical bolometry and conclude that it has a negligible effect when compared to direct photothermal tensioning. We have described the results of this analysis in the manuscript and provide the details in the SI.*
 - 4. Fourth, in response to the concerns raised by Reviewer #3 about bandwidth, we have analyzed the bandwidth limitations of the bolometer and made the distinction between thermal response time, mechanical bandwidth, and the bolometer bandwidth clearer. For this work, we used frequency modulation, and consequently the bolometer bandwidth is not limited by the mechanical linewidth, as described by Albrecht et al.⁸ in their paper “Frequency modulation detection using high-Q cantilevers for enhanced force microscope sensitivity.” Unlike the mechanical amplitude, the resonance frequency responds instantaneously to external forces. By actively feeding back on deviations in phase, frequency modulation detection can respond faster than the mechanical linewidth. We have calculated the thermal response bandwidth and mechanical linewidth for all devices and added these values to the SI. Direct measurements of the bolometer bandwidth agree with bandwidth predicted by the thermal response and not the mechanical linewidth. In some case, the measured device response time was two orders of magnitude larger than that predicted from the mechanical linewidth.*
- “- Study of the spectral response of the bolometers: there exist an expansive literature on interference effects to which suspended graphene NEMS are subjected. This will affect the response of the device which will certainly not be flat band. A spectral study of the device is therefore necessary.

- Effect of non-coherent wide-spectrum radiation on the response of the device: subjecting the device to a non-laser light illumination, and studying the effect of multi-spectral illumination on the device is also necessary to claim practical significance.”

We thank the reviewer for this suggestion. To consider how the absorption changes in a suspended graphene cavity, we have added a cavity modeling theory section to the SI. This analysis provides a framework to understand how the cavity depth can be used to selectively enhance the absorption at different wavelengths. In-depth studies of the spectral response and incoherent light absorption are not standard for graphene bolometer publications^{1,11-13} or for bolometer publications in general, especially those involving a proof of principle of a novel approach, such as ours. Therefore, we respectfully disagree with the reviewer that spectral studies are necessary for this publication. Such studies would involve significant experimental work not currently possible with our current optical apparatus, but could make interesting future work. We hope that our expanded theoretical analysis on how cavity effects are expected to modulate the spectral response will satisfy Reviewer #3.

- “- The DC voltage can be used to preferentially enhance the bolometer’s response to a certain wavelength once the spectral response of the Si/SiOx/vacuum/graphene is taken into account. A detailed experimental study of the effects of DC voltage would therefore be of interest.”

We thank the reviewer for the comment. While a DC bias could, in principle, modulate or even tune the spectral response, the DC bias will also increase the mechanical stress in the device, which will in turn negatively affect the noise-equivalent power of the device. We expect this trend from our mechanical model which predicts that the frequency responsivity is inversely proportional to the initial stress of the device. In the SI we measure the frequency responsivity (at fixed wavelength) as the DC voltage was increased and found that as the DC voltage was increased, the frequency responsivity was subsequently decreased. Because a high DC voltage increases the initial stress of the graphene it is a counterproductive technique to enhance the absorption. We have expanded the modeling of the electrostatic stress in the SI.

- “- The authors do not attempt to fit the data (plots 2.c, 3.b, and 3.d) to their model, and extract useful parameters. Instead they rely on textbook values in SI.”

We thank the reviewer for this comment. The data in these plots (i.e. of the frequency responsivity, bandwidth, and noise-equivalent power) could in principle be used to extract useful parameters. For example, the frequency responsivity could be used to obtain the Young’s modulus or the membrane stress. However, the main purpose of our modeling is not extract known physical parameters, but rather to explore the geometric dependence of the trampoline. We now fit the frequency responsivity, bandwidth, and noise-equivalent power to the geometric dependence on the tether width, and report the associated R-values to confirm their linear or inverse dependence predicted by our modeling. To extract parameters, we have added modeling of the resonance frequency vs. DC voltage sweep for a graphene drumhead to the SI and manuscript. This analysis allows us to extract useful parameters including the mass density, initial stress, and elastic modulus, which in turn allow us to independently predict and calculate the graphene heat capacity, thermal resistance, graphene optical absorption, and the frequency

responsivity of bolometer. We then use these values to estimate the frequency responsivity and find that it agrees with the independent direct measurement of the frequency responsivity within a factor of ~10%.

- “- The photothermal back-action is well known (Barton et al., Metzger et al.), and should be quantified and modeled by looking at changes in linewidth.”

We thank the reviewer for this suggestion. We have added a section to the supplementary information to estimate the strength of photothermal back-action with regard to bolometry. We find that this effect is not significant for our measurements and it is at least two to four orders of magnitude weaker than photothermal tensioning. Furthermore, our measurements use nanowatts of power to characterize the bolometer performance, while the work done in (Barton et al.¹⁴) used milliwatts of power to see significant photothermal back-action, and Barton et al. employed a backside mirror to enhance back-reflections, a feature our devices do not contain.

- “- Most importantly, either the authors do not explain this well enough in the text or they have committed a category mistake: In the model they seem to assume that the thermal time constant is the same as the device response time, i.e. BW. That is incorrect. The thermal time constant is set by the thermal conductivity, capacity, and interface resistance (Dolleman et al.) whereas the response of a resonant bolometer, i.e. its BW, is that of the mechanical linewidth (to a first order approximation), it is not surprising then that the BW measured using the PLL seems to be equal to the mechanical linewidth of the resonator. Thus equating the mechanical and thermal time constant is a fundamental error.”

Reviewer #3 states that the thermal time constant is defined by thermal resistance and heat capacity. We agree 100%. Our original manuscript and original SI included several instances and sections (e.g. the introduction) discussing this fact about the thermal time constant, namely that $\tau_T = R_T C$, where τ_T is the thermal response time, and R_T and C are the thermal resistance and capacity, respectively. We've modified the wording in the manuscript to clarify references to the thermal response time.

*Review #3 then states that this thermal time constant and the response time of our nanomechanical bolometer are not the same, because the nanomechanical resonance response is set by the resonance linewidth. We'd like to respectfully point out that Reviewer #3's statement is incorrect. Resonant sensing speed is set by the linewidth, as Reviewer #3 states, but only if the sensing is performed with amplitude modulation (i.e. slope detection). However, we did not use amplitude modulation in our measurements. We used frequency modulation. As pointed out by D. Rugar's group⁸ (“Frequency modulation detection using high-Q cantilevers for enhanced force microscope sensitivity”. *J. Appl. Phys.* **69**, 668–673 (1991)), a highly cited paper, one of the key advantages of frequency modulation detection is that the bandwidth is not limited by the mechanical linewidth. Instead, the upper limit of the bandwidth is set by the mechanical resonance frequency, which in our case is ~10 MHz. But the measured bandwidth of our nanomechanical bolometer is far below the resonance frequency, even for the high bandwidth of circular drumhead resonators, because other mechanisms limit the response speed. In our system, the bandwidth is limited by either the PLL speed (~50 kHz) or the thermal response time, whichever is slower. In cases where the PLL is not the limiting factor, we infer the thermal*

response time (that is, the device bandwidth) directly from the frequency response of the mechanical resonant frequency-shift responsivity. In cases where the PLL was a limiting factor, we infer the thermal response time from the off-resonant thermomechanical amplitude response. We provide full and expanded discussion of these measurements and the modeling in the SI, hoping that the additional information will help the reader better understand the device bandwidth.

In cases where we did directly measure the response bandwidth using both the resonant and off-resonant methods, we compare the respective values and find good agreement, which we show in the SI. For the sake of comparison, we also estimate a bandwidth from the mechanical linewidth (i.e. the bandwidth in amplitude-modulation-based sensing) and find that the linewidth does not predict the bandwidth of the device, as expected with frequency modulation. We have added the response bandwidth and the resonance linewidth to Table 1 of the SI so that these metrics can be directly compared.

Review #3 does say that we perhaps didn't explain the bandwidth well enough. We agree, and this lack of explanation likely contributed to Reviewer #3's confusion. To correct this, we have added more discussion about bandwidth including what limits the bandwidth and how its determined by the thermal response time, or in practice, the PLL. The SI now has a dedicated section on bandwidth measurements and comparisons.

- “- The authors have completely ignored the mechanics (dynamics) of their resonator in their model, a complete model should account for such effects.”

The mechanical modeling was included in the original SI, and the key points of the model were included in the main, but they were perhaps not so clear. In the updated version of the manuscript we have significantly expanded our mechanical modeling and organized the modeling for clarity. These are included in the thermomechanical model which is used to write an expression for the frequency responsivity and a new electromechanical experiment and model fit which is used to extract useful mechanical parameters.

The “Thermomechanical Modeling” section in the SI adds thermomechanical stress to the equation for the resonance frequency for a circular membrane to predict how a rise in temperature in the suspended membrane shifts the resonant frequency. The “Electromechanics” section in the SI uses a three-parameter fit to a DC voltage vs. resonance frequency sweep to extract the initial stress, mass density, and modulus. By using this model and experiment to extract these mechanical parameters we predict the frequency responsivity to within 10% of the directly measured value. The “Bandwidth Measurements” section of the SI provides a theoretical connection between the off-resonant amplitude and resonant frequency response. The “Effects of the Optical Cavity” section in the SI develops the photothermal mechanical back-action. We believe these updates will provide sufficient mechanical modeling for publication.

References:

1. Efetov, D. K. *et al.* Fast thermal relaxation in cavity-coupled graphene bolometers with a Johnson noise read-out. *Nat. Nanotechnol.* **13**, 797–801 (2018).
2. Steele, G. a *et al.* Strong coupling between single-electron tunneling and nanomechanical motion.

- Science* **325**, 1103–7 (2009).
3. Dominguez-Medina, S. *et al.* Neutral mass spectrometry of virus capsids above 100 megadaltons with nanomechanical resonators. *Science* **362**, 918–922 (2018).
 4. Chaste, J. *et al.* A nanomechanical mass sensor with yoctogram resolution. *Nat. Nanotechnol.* **7**, 301–304 (2012).
 5. Jensen, K., Kim, K. & Zettl, a. An atomic-resolution nanomechanical mass sensor. *Nat. Nanotechnol.* **3**, 533–7 (2008).
 6. Yang, H. & Rebeiz, G. M. Sub-10-pW/Hz0.5 Uncooled Micro-Bolometer With a Vacuum Micro-Package. *IEEE Trans. Microw. Theory Tech.* **64**, 2129–2136 (2016).
 7. Skidmore, G. D., Han, C. J. & Li, C. Uncooled microbolometers at DRS and elsewhere through 2013. *Proc. SPIE* **9100**, 910003 (2014).
 8. Albrecht, T. R., Grütter, P., Horne, D. & Rugar, D. Frequency modulation detection using high-Q cantilevers for enhanced force microscope sensitivity. *J. Appl. Phys.* **69**, 668–673 (1991).
 9. Laurent, L., Yon, J.-J., Moulet, J.-S., Roukes, M. & Duraffourg, L. 12- μ m-Pitch Electromechanical Resonator for Thermal Sensing. *Phys. Rev. Appl.* **9**, 024016 (2018).
 10. Zhang, X. C., Myers, E. B., Sader, J. E. & Roukes, M. L. Nanomechanical torsional resonators for frequency-shift infrared thermal sensing. *Nano Lett.* **13**, (2013).
 11. Han, Q. *et al.* Highly sensitive hot electron bolometer based on disordered graphene. *Sci. Rep.* **3**, 3533 (2013).
 12. Yan, J. *et al.* Dual-gated bilayer graphene hot-electron bolometer. *Nat. Nanotechnol.* **7**, 472–478 (2012).
 13. El Fatimy, A. *et al.* Epitaxial graphene quantum dots for high-performance terahertz bolometers. *Nat. Nanotechnol.* **11**, 335–338 (2016).
 14. Barton, R. A. *et al.* Photothermal Self-Oscillation and Laser Cooling of Graphene Optomechanical Systems. *Nano Lett.* **12**, 4681–4686 (2012).

REVIEWERS' COMMENTS:

Reviewer #1 (Remarks to the Author):

After considering the authors' reply to the referee comments and the substantial improvements of the manuscript and supplemental material, I believe that this work worth to be published in Nature Communications.

Reviewer #2 (Remarks to the Author):

The authors have taken the effort to think about and respond adequately to the concerns raised. Overall, the manuscript is suitable for publication at this stage.

Reviewer #3 (Remarks to the Author):

The authors of the manuscript "A fast, sensitive, room-temperature graphene nanomechanical bolometer" have provided a highly revised version of their initial manuscript.

In the new manuscript, the authors have addressed most of the comments made by the referees, and provided valuable clarification to the questions posed.

In the new manuscript, several sections have been added to the supplementary information, mainly concerning optical cavity effects including back-action, and device physical parameters.

The authors have equally improved the text so that several previous sticking points are now easily understandable.

I recommend publication of the manuscript, after the following minor modifications:

- Some typos can be found throughout the text, example line 205 of the manuscript. Please take one more look.
- In line 94 of the main manuscript, they state "below the onset of nonlinearity". It seems from the context, they mean "bistability". Onset of nonlinearity may be defined as the 1db compression point (H. W. C. Postma, I. Kozinsky, A. Husain, and M. L. Roukes, Appl. Phys. Lett. 86, 223105, 2005). No need for a reply, correct accordingly.

REVIEWERS' COMMENTS:

Reviewer #1 (Remarks to the Author):

After considering the authors' reply to the referee comments and the substantial improvements of the manuscript and supplemental material, I believe that this work worth to be published in Nature Communications.

We thank reviewer #1 for supporting the publication of this work.

Reviewer #2 (Remarks to the Author):

The authors have taken the effort to think about and respond adequately to the concerns raised. Overall, the manuscript is suitable for publication at this stage.

We thank reviewer #2 for supporting the publication of this work.

Reviewer #3 (Remarks to the Author):

The authors of the manuscript "A fast, sensitive, room-temperature graphene nanomechanical bolometer" have provided a highly revised version of their initial manuscript. In the new manuscript, the authors have addressed most of the comments made by the referees, and provided valuable clarification to the questions posed. In the new manuscript, several sections have been added to the supplementary information, mainly concerning optical cavity effects including back-action, and device physical parameters. The authors have equally improved the text so that several previous sticking points are now easily understandable.

We thank reviewer #3 for supporting the publication of this work.

I recommend publication of the manuscript, after the following minor modifications:

- Some typos can be found throughout the text, example line 205 of the manuscript. Please take one more look.

We have fixed this typo by changing "or model" to "our model" and scanned the document for other typos.

- In line 94 of the main manuscript, they state "below the onset of nonlinearity". It seems from the context, they mean "bistability". Onset of nonlinearity may be defined as the 1db compression point (H. W. C. Postma, I. Kozinsky, A. Husain, and M. L. Roukes, Appl. Phys. Lett. 86, 223105, 2005). No need for a reply, correct accordingly.

We have changed this sentence according the recommendation.